# Barriers in the access, diagnosis and treatment completion for tuberculosis patients in central and western Nepal: A qualitative study among patients, community members and health care workers

Sujan Babu Marahatta[1], Rajesh Kumar Yadav[1], Deena Giri[1], Sarina Lama[1], Komal Raj Rijal[2], Shiva Raj Mishra[3], Ashish Shrestha[4], Pramod Raj Bhattrai[4], Roshan Kumar Mahato[5], Bipin Adhikari[6,7] *

1 Manmohan Memorial Institute of Health Sciences, Soaltee mode, Kathmandu, Nepal, 2 Central Department of Microbiology, Tribhuwan University, Kirtipur, Kathmandu, Nepal, 3 University of Queensland, Queensland, Australia, 4 National Tuberculosis Centre, Bhaktapur, Nepal, 5 Dhulikhel Hospital, Dhulikhel, Nepal, 6 Nepal Community Health and Development Centre, Kathmandu, Nepal, 7 Centre for Tropical Medicine and Global Health, Nuffield Department of Medicine, University of Oxford, Oxford, England, United Kingdom

* biopion@gmail.com

**Data Availability Statement:** Data cannot be shared publicly because of the nature of the data

## Abstract

### Background

Nepal has achieved a significant reduction of TB incidence over the past decades. Nevertheless, TB patients continue to experience barriers in access, diagnosis and completion of the treatment. The main objective of this study was to explore the factors affecting the access to the health services, diagnosis and the treatment completion for TB patients in central and western Nepal.

### Methods

Data were collected using in-depth interviews (IDI) with the TB patients (n = 4); Focus Group Discussions (FGDs) with TB suspected patients (n = 16); Semi Strucutred Interviews (SSIs) with health workers (n = 24) and traditional healers (n = 2); and FGDs with community members (n = 8). All data were audio recorded, transcribed and translated to English. All transcriptions underwent thematic analysis using qualitative data analysis software: Atlas.ti.

### Results

Barriers to access to the health centre were the long distance, poor road conditions, and costs associated with travelling. In addition, lack of awareness of TB and its consequences, and the belief, prompted many respondents to visit traditional healers. Early diagnosis of TB was hindered by lack of trained health personnel to use the equipment, lack of equipment and irregular presence of health workers. Additional barriers that impeded the adherence

being qualitative that contains personal quotes and clues to where the study occurred and can be potentially identifiable. However, data is available on request to the chair of the research department (E-mail: drdharmakhanal@gmail.com) complying with the data access policy outlined by Institutional Review Committee (IRC) of Manmohan Memorial Institute of Health Sciences (http://www.mmihs.edu.np/irc.php).

**Funding:** This study was funded by grants from the WHO-TDR, the Special Programme for Research and Training in Tropical Diseases. ID number: 178861. The funders had no role in study design, data collection and analysis, decision to publish, or preparation of the manuscript.

**Competing interests:** The authors have declared that no competing interests exist.

**Abbreviations:** DOTS, Directly Observed Treatment Short-course; DTLS, District TB and Leprosy Supervisor; DTLO, District TB Leprosy Officer; DPHO, District Public Health Office; EDR, Eastern Development Region; FWDR, Far Western Development Region; FGDs, Focus Group Discussions; HIV/AIDS, Human Immunodeficiency Virus/Acquired Immune Deficiency Syndrome; HP, Health Posts; IDIs, In Depth Interviews; MDR, Mid Western Region; NTC, National Tuberculosis Centre; SSIs, Semi-Strucutred Interviews; TB, Tuberculosis; WHO, World Health Organization; WDR, Western Development Region.

and treatment completion were the need to visit health centre daily for DOTS treatment and associated constraints, complex treatment regimen, and the stigma.

## Conclusions

Barriers embedded in health services and care seekers' characteristics can be dealt by strengthening the peripheral health services. A continuous availability of (trained) human resources and equipment for diagnosis is critical. As well as increasing the awareness and collaborating with the traditional healers, health services utilization can be enhanced by compensating the costs associated with it, including the modification in current DOTS strategy by providing medicine for a longer term under the supervision of a family member, peer or a community volunteer.

## Introduction

Tuberculosis (TB) is one of the top 10 causes of death globally and led to 1.6 million deaths in 2017 [1]. Despite the availability of efficacious treatment for TB, high morbidity and mortality affect low and middle-income countries disproportionately [1, 2]. The highest number of new TB cases occurred in the South East Asia and Western Pacific regions, which included nearly two-thirds of the new cases, followed by the African region, with one-fourth of new cases. A very high morbidity (95% of TB cases) with mortality upto 98% have been reported from South East Asia and the Africa [3]. Exacerbated by poverty, poor public health systems, and increasing HIV/AIDS prevalence, TB continues to be a persistent challenge for global health and development [4]. Poverty further accentuates their vulnerability to poor access, diagnosis and treatment completion [5].

Globally in 2017, 10 million people developed TB disease, of which 5.8 million (58%) were men, 3.2 (32%) million women and 1 million (10%) children [6]. Almost half of the population (45%) in Nepal is infected with TB and 60% of the infected population is from the productive age group [7]. In addition, 44,000 people develop active TB each year and more than half (22,500) of these active TB have the infectious type of pulmonary TB. Annually, between 5,000 to 7,000 people die of TB in Nepal [8].

One of the key challenges in mitigating the mortality associated with TB is due to the barriers in getting access to health services, diagnosis and the adherence to treatment [9]. Various factors such as economic barriers [10, 11] including the integration of TB services with health care system, number of visits, cost of seeking care plays a critical role [12] in determining the access (of health services), diagnosis (of TB) and adherence of treatment regimen [13]. These barriers are further augmented by the complex geographical terrain and the factors embedded in social and cultural context [14].

In Nepal, patients experienced more than five weeks of delay in diagnosis and the treatment of pulmonary tuberculosis [15]. Delays in access and diagnosis together with the poor adherence poses serious health and economic implications for both the individual, family members and the community [14, 15]. For instance, poor access and delayed diagnosis can lead to severe form of the disease, spread of the disease among the family members and the community members [16–18]. In addition, poor adherence to treatment can lead to the development of drug-resistant TB, the persistence of tuberculosis which may remain as a reservoir of infection in the community [19].

Previous studies from Nepal have explored the delays [15] and barriers in accessing the TB treatment in Southern [20] and Eastern Nepal [21] among the TB patients. Nevertheless, there is a paucity of evidence on an integral understanding of how these barriers affect access, diagnosis and treatment completion in various geographic regions based on the opinions of the health care providers, patients and community members. The main objective of this study was to explore the factors affecting the access to health services, diagnosis and the treatment completion for TB patients in central and western Nepal.

## Materials and methods

### Setting

Nepal is composed of three ecological zones that comprises upper Himalayan region *(Himal)*, middle mountains *(Pahad)* and the southern plains *(Terai)*. This study was carried out in three various geographic regions of Nepal (Fig 1). A total of six districts were chosen based on the literature review that informed the paucities of evidence on barriers of access, diagnosis and treatment completion on TB in central and western Nepal [14]. Two districts in each of these three regions constituted a total of six districts: *Tanahuh, Kaski, Parsa, Nawalparasi, Mustang and Kathmandu.*

### Study design and participants

This was a qualitative study conducted in six districts in central and western Nepal and follows a standard qualitative research methodology [22]. The study utilized a range of data collection techniques based on the phenomenological approach in qualitative research [23]: Semi-Structured interviews (SSI), Focused Group Discussions (FGDs), and In-Depth Interviews (IDIs) among a range of participants. Twelve FGDs, two each in a district were conducted among 69 community participants (Table 1). Twenty-one SSIs were conducted among government health service providers at various levels (Table 2). Three SSIs were conducted among private health service providers (Table 3) and two among traditional health service providers (Table 4). In-depth interviews were conducted with four patients each in a unique category of treatment (Table 5) and 16 FGDs (eight each in two major TB centres) were conducted among TB suspected patients (Table 6). All participants were purposively selected for the study based

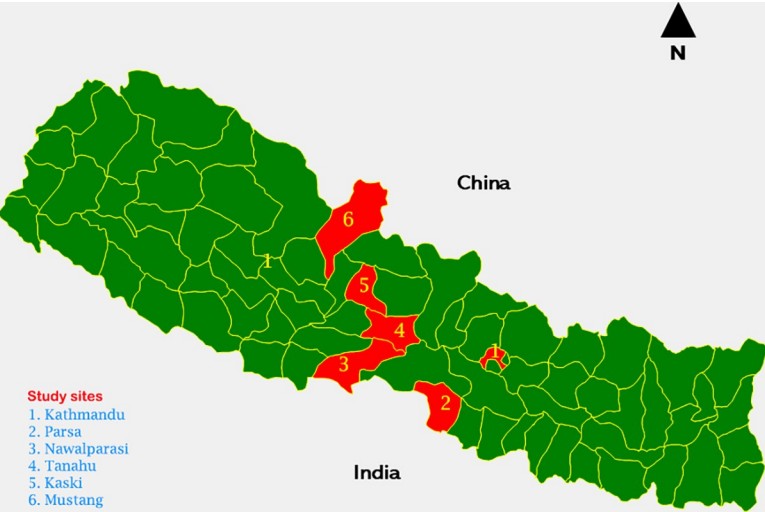

**Fig 1. Study sites by geographic region and districts.**

Study sites
1. Kathmandu
2. Parsa
3. Nawalparasi
4. Tanahu
5. Kaski
6. Mustang

**Table 1. Total number of FGD participants from communities.**

| District | FGDs | Village Development Committee | Sex | Total number of participants |
|---|---|---|---|---|
| Mustang | FGD_M_1 | Lete-1 | Males = 2 and Females = 3 | 11 |
| | FGD_M_2 | Lete-2 | Males = 2 and Females = 4 | |
| Kaski | FGD_K_1 | Lumle-1 | Males = 4 and Females = 1 | 11 |
| | FGD_K_2 | Lumle-2 | Males = 3 and Females = 3 | |
| Tanahun | FGD_T_1 | Dulegaunda-1 | Males = 3 and Females = 2 | 10 |
| | FGD_T_2 | Dulegaunda-2 | Males = 2 and Females = 3 | |
| Nawalparasi | FGD_N_1 | Makar-1 | Males = 4 and Females = 3 | 15 |
| | FGD_N_2 | Makar-2 | Males = 5 and Females = 3 | |
| Parsa | FGD_P_1 | Pokhariya-1 | Males = 4 and Females = 3 | 12 |
| | FGD_P_2 | Pokhariya-1 | Males = 3 and Females = 2 | |
| Kathmandu | FGD_KU_1 | Suichatar-1 | Males = 3 and Females = 2 | 10 |
| | FGD_KU_2 | Suichatar-2 | Males = 3 and Females = 2 | |
| Total | | | | 69 |

on their likelihood to provide information relevant to the research question. A total of six invited participants (TB suspected patients) could not participate because of their busy appointments at hospital.

The field workers led by RKY and supervised by SBM at first contacted the respective authorities at central, and district level with an ethical approval letter from Nepal Health Research Council and Institutional Review Committee (IRC) of Manmohan Memorial Institute of Health Sciences. National Tuberculosis Centre (NTC) of Nepal coordinated and facilitated the the study at various levels. A sensitization meeting was held at each level prior to the data collection among the authorities. At the district level, district public health officer coordinated and facilitated the study including identification of the respondents for FGDs, SSIs and IDIs.

Prospective participants of the study were at first briefed about the study, its procedure and rationale and were encouraged to ask questions. All participants who agreed to participate were further detailed about the written consent procedure and were explained to sign the written informed consent. Participants were explained that they could opt out of the study anytime during the interviews/discussions without requiring to provide a reason. FGDs with community participants took place at the village development committee's meeting room. SSIs with various level of government health service providers were arranged by appointment and took place at their offices. SSIs with private sector health service providers took place at their suggested locations, either at the hospital/DOTS centre or other locations, for example at the garden near the hospital. SSIs with traditional healers took place by appointment at their houses. IDIs with various categories of patients were conducted at the quiet location in the hospital. A total of 16 FGDs consisting of three to eight suspected TB patients were conducted at either National Tuberculosis Centre, Bhaktapur or at Genetop TB centre, Kalimati. None of the participants was provided with monetary incentives.

## Data collection

All interviews from SSIs and IDIs together with the FGDs were conducted face to face and were audio recorded by two interviewers under direct supervision of RKY who had extensive training in qualitative research methods. Three field researchers (RKY, DG and SL, graduate in public health and qualitative research) were trained for five days on the procedure and the

**Table 2. Semi-structured interview with various level of government health service providers.**

| SSI with Government health service Providers | District Name | Level of the health service provider | Age in years | Sex |
|---|---|---|---|---|
| SSI_GHSP_1 | Mustang | DTLO | 20 | Male |
| SSI_GHSP_2 | Mustang, Jharkot HP | DOTS focal person | 28 | Male |
| SSI_GHSP_3 | Kaski | DTLO | 51 | Male |
| SSI_GHSP_4 | Kaski, DPHO | DOTS focal person | 38 | Female |
| SSI_GHSP_5 | Tanhun | DTLO | 52 | Male |
| SSI_GHSP_6 | Tanhun, Aabookhairani HP | DOTS focal person | 37 | Male |
| SSI_GHSP_7 | Nawalparasi | DTLO | 55 | Male |
| SSI_GHSP_8 | Nawalparashi, Makar HP | DOTS focal person | 35 | Female |
| SSI_GHSP_9 | Parsa | DTLO | 55 | Male |
| SSI_GHSP_10 | Parsa, Birgunj HP | DOTS focal person | 34 | Female |
| SSI_GHSP_11 | Kathmandu | DTLO | 48 | Female |
| SSI_GHSP_12 | Kathmandu, Gokarn PHC | DOTS focal person | 33 | Male |
| SSI_GHSP_13 | EDR | RTLO | 53 | Male |
| SSI_GHSP_14 | WDR | RTLO | 55 | Male |
| SSI_GHSP_15 | MWDR | RTLO | 48 | Male |
| SSI_GHSP_16 | FWDR | RTLO | 38 | Male |
| SSI_GHSP_17 | National Level | NTC | 35 | Male |
| SSI_GHSP_18 | National Level | NTC | 55 | Male |
| SSI_GHSP_19 | National Level | NTC | 33 | Male |
| SSI_GHSP_20 | National Level | NTC | 48 | Male |
| SSI_GHSP_21 | National Level | Genetup | 55 | Female |

SSI_GHSP = Semi-structured interview with government health service providers

methods of data collection by SBM and BA who have expertise in qualitative social science research. Any ambiguities in data collection guides were corrected through discussions among the researchers first.

Data collection guides were allowed to be flexible to ensure the exploration of important themes as it emerged. During the data collection, an assistant to interviewer collected the notes about the atmosphere, the general tone of the interviews and utterances. The sample size for this study including the number of FGDs, SSIs, IDIs and the participants were considered adequate based on the theoretical saturation, whereby no novel findings emerged from the subsequent interviews [24, 25].

## Data analysis

All audio recorded interviews and discussions together with the field notes taken during the interviews were transcribed and translated in the English language. A pre-designed codebook guided the initial data analysis of the themes using Atlas.ti. Three researchers (RKY, DG and

**Table 3. Semi Structure Interview with private sector health service provider.**

| SSI with the private sector | District Name | Level of the health service provider | Age | Sex |
|---|---|---|---|---|
| SSI_PS_1 | Manipal Medical College, Private | DOTS focal person | 42 | Female |
| SSI_PS_2 | Narhari Medical, Private | DOTS focal person | 58 | Male |
| SSI_PS_3 | Kathmandu Medical College, Private | DOTS focal person | 51 | Female |

SSI_PS = Semi-structured interview with the private sector

**Table 4. Semi Structure Interview for a traditional health service provider.**

| SSI with traditional healers | District Name | Age | Sex |
|---|---|---|---|
| SSI_TH_1 | Bagar, Kaski | 65 | Male |
| SSI_TH_2 | Hemja, Kaski | 52 | Female |

SSI_TH = Semi-structured interview with traditional healers

SL) coded the data independently. A thematic analysis was conducted using pre-established codes (deductive approach) and additional codes were created for the emerging themes (inductive approach) [26]. At first, codes were discussed among three researchers who coded the data, and later discussed with SBM and BA for consistency and validity. Together with the interviewers, members of the research team reviewed the data analysis; and disagreements between themes and its interpretations were discussed and resolved after seeking an alternative opinion from a member who was not involved in the data analysis. Both minor and major themes/sub-themes regardless of their frequency but relevant to research question were included and are presented in this study. The final themes relevant to the research question are broadly categorized into A: Health system level barriers and B: Care seekers' socio-demographic barriers (Fig 2).

## Ethics approval

Ethical approval for this study was obtained from the Nepal Health Research Council (Reference number:261/2015) and Institutional Review Committee (IRC) of Manmohan Memorial Institute of Health Sciences. Written informed consent was obtained from each participant before the interviews/discussions.

## Results

### A: Health system level barrier

Nepal has a unique landscape with regions divided between the northern Himalayas bordering China, mid mountains and the southern plains bordering India (Fig 1). Nepal has established a good network of peripheral health structures over the decades which serves the rural population. However, this study identified myriad of factors that affected the health service utilization leading to a neglect of one of the killer disease: Tuberculosis (Fig 2). Below, we outline each of these factors affecting health service utilization.

**Availability of human resources at the peripheral health structure.** Scarcity of staff at peripheral health structures has been a major problem in Nepal and has been echoed by respondents in all six districts. Irregularity of staff's attendance, official hours and sometimes lack of staff for many days were major impediments for patients. Most of the participants

**Table 5. In-depth interview for different treatment categories of TB Patients.**

| IDI | Treatment categories of TB Patients | Age in years | Sex |
|---|---|---|---|
| IDI_TBP_1 | Treatment Successfully | 32 | Male |
| IDI_TBP_2 | Treatment Failed | 49 | Male |
| IDI_TBP_3 | Loss to follow up(Defaulted) | 23 | Male |
| IDI_TBP_4 | Died(relatives of those TB patient died) | 53 | Female |

IDI_TBP = In-depth interview with TB patients

**Table 6. FGD for TB suspected patients.**

| Place of FGD | Sex | Total number of participants |
|---|---|---|
| **National Tuberculosis Centre (NTC), Bhaktapur Thimi** | | |
| FGD_NTC-1 | Male = 5 and Female = 3 | 8 |
| FGD_NTC-2 | Males = 4 and Females = 3 | 7 |
| FGD_NTC-3 | Males = 5 and Females = 1 | 6 |
| FGD_NTC-4 | Males = 4 and Females = 4 | 8 |
| FGD_NTC-5 | Males = 4 and Females = 2 | 6 |
| FGD_NTC-6 | Males = 6 and Females = 2 | 8 |
| FGD_NTC-7 | Males = 5 and Females = 1 | 6 |
| FGD_NTC-8 | Males = 6 and Females = 1 | 7 |
| **Genetup, TB diagnostic centre, Kalimati** | | |
| FGD_Genetup_1 | Male = 4 and Female = 3 | 7 |
| FGD_Genetup_2 | Males = 6 and Females = 2 | 8 |
| FGD_Genetup_3 | Males = 5 and Females = 1 | 6 |
| FGD_Genetup_4 | Males = 5 and Females = 1 | 6 |
| FGD_Genetup_5 | Males = 6 and Females = 2 | 8 |
| FGD_Genetup_6 | Males = 4 and Females = 1 | 5 |
| FGD_Genetup_7 | Males = 3 and Females = 1 | 4 |
| FGD_Genetup_8 | Males = 2 and Females = 1 | 3 |

shared that staff were unavailable even during office hours, health facility did not follow office schedule and participants could not find health workers in the health facility for many days because of which patients were not able to receive services.

Also, the lack of staff who could detect, treat and dispense medicine was a major problem resulting in a missed opportunity for diagnosis, delayed initiation of the medicine, inadequate counselling and poor adherence to medication. Some patients' disappointments were further aggravated when they had to near the cost of attending a higher health centres to be diagnosed as TB.

*"Doctors in the health facility are not regular. So, people are examined by the health assistants and Auxiliary Nurse Midwife. . . we have the case that they provided wrong medicine due to which the patient's problems got complicated." Few years ago, one of my neighbour travelled to Pokhara and was found to have TB. This could easily have been done here"*

*(FGD with six community members, Mustang, Lete)*

Lack of human resources and the constant presence in peripheral health structures are a nationwide problem. While lack of human resources in peripheral health structures invariably affects all diseases and conditions, inefficient TB case detection may also have stemmed from the government policies (citing the budgetary constraints) that withdrew the incentives for sputum collection for community health workers.

**Shortage of medications and lab resources.** Respondents were often dissapointed by the lack of medications at the health centres. Upon asking the health workers/service providers, the reasons were often attributed to a lack of supply from the higher centres. Even lack of syringes and distilled water were sometimes the reasons for deferring patients to other centres, particularly for those patients who were undergoing treatment for category II Tuberculosis.

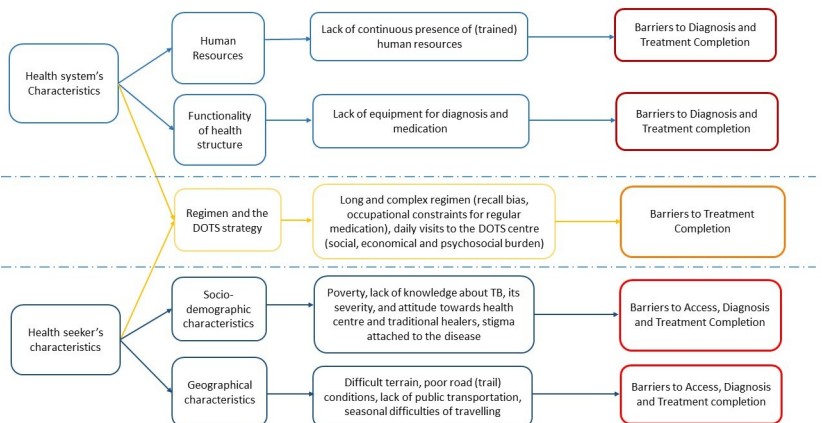

**Fig 2. Factors affecting the barriers in access, diagnosis and treatment completion for tuberculosis patients.**

*". . .we have to give streptomycin for category 2 patient. For this, sometime there will be no syringe and sometimes there is a lack of distilled water and sometimes there are no medicines. We cannot give injectable to the patient without these three things. . .. . ..Also, patients are referred from Kathmandu but medicines are not available in our health facility on time".*

*(SSI with DOTS focal person, Parsa)*

Health centres were often under-resourced by the necessary lab staff and the reagents to detect the new case of TB. Often the patients were referred to regional TB centres for the diagnosis based on the empirical judgement on signs and symptoms. Some patients were often challenged with antibiotics and if not resolved were sent for sputum test at the regional centres. Referral for these reasons also led patients to incur additional direct and indirect costs of attending the regional health centres.

*"We refer to regional TB centre only on the basis of sign and symptoms because nothing is here like sputum test, Mantoux test, lab facility as a base for TB test. We are only doing on the basis of symptoms".*

*(DTLO, Kaski)*

While the diagnosis was often constrained in peripheral health structures, the transportation of samples and receiving back the diagnosis were also reported to be inefficient. Such a delay further engendered the increase in severity of disease and transmission among the family members and neighbours.

**Location of health centres.** Both patients and health staff acknowledged the lobbying and politicization of establishing a health centre for the villages. Often these health centres are established at a location convenient and near to the residence of political leaders and the access issues to such health centres by the majority of the population are often underscored. In addition, villages in mountains where settlements are sparse and are spread in a large mountainous area, attending health centres can become a significant problem.

*"Health facilities have been built near the house of powerful political leaders. This is convenient for them [political leaders], but not for the [common] people. In this village, look where the population is mostly living and how far they have to travel to reach".*

*(IDI, TB patient, Kaski)*

Apart from issues of accessibility to health centres, some patients who are in treatment often fail to adhere to medication due to the migration of families. In general, people who work in daily wages, often migrate for work in different areas and current policies of DOTS restrain health staff from providing medication for home. Often these group of people are double burdened by the economic vulnerability, need for migration and relapses of TB.

**Patient-Provider Relationship.** Lack of support and compassion to patients can affect the patient's trust and likelihood to attend the health centre. Lack of adequate time to discuss the health condition and the poor comprehension about the disease often deterred patients to attend the health centre. Few participants expressed the short and unprofessional behaviour from health centre staff and it discouraged their future attendance.

## B. Geographical and socio-demographic barriers

**Geographical barrier.** One of the most pronounced factor from all the respondents at all districts was the distance to the health centre. Coupled with the geographical barriers to access, for instance, uphill route, forested path, and rivers/springs in the path towards health centres with the long distance mostly discouraged patients to access the health services. Respondents also stressed that attending health centres for women, elderly and severely sick person further adds impediment to accessing the health services. In addition, in rural villages in Nepal, with the amount of work related to subsistence farming, cattle rearing, collecting water, and household chores including cooking, patients felt almost impossible to walk those distances to reach the health centres.

*"People have to walk 7–8 km distance to visit a health facility. In addition, the roads are not in plain, they have to cross the hills, forests, rivers and springs. How would they walk without a company to the health centre? We have so much [agricultural, work related to cattle] work at home, how can we spare so much time to visit the health centre? So it is difficult for us to go for treatment at the health centre".*

*(FGD with five community members at Tanahun)*

**Lack of transportation facilities to reach TB services.** Geographical barriers due to difficult terrain have been further complicated by the lack of roads for vehicles and means of public transportation. Even seemingly, accessible districts in the southern Terrain were more remote due to lack of roads and public transport than the remotest districts of Nepal such as *Humla* and *Jumla*, which are in the far north-western region of Nepal.

*"Parsa district has many remote places from where it is difficult to visit a health facility. Parsa is a Terai district but this place is more remote than Humla and Jumla [Remote districts of Nepal]. There are 3 VDCs in Parsa where there is no transportation facility".*

*(IDI, DOTS Focal person, Parsa)*

A significant number of participants during FGDs and individual interviews expressed that because of the lack of transportation facilities people were not able to utilize health service. Mostly, means of transportations were not available or even if they were available, people could not get a vehicle at the time of need. While there was consistency among respondents about this problem, undoubtedly, this had an impact on the maintenance of health services.

For instance, the lack of transportation facilities constantly impeded the supply of medicines and equipment.

*". . ...In my opinion, the lack of availability of medicine is a major barrier. Medicines are not available on time due to transportation problems".*

*(IDI, Health staff, Mustang)*

While the distance and the availability of transportation affected both access and the chances of early case detection, prominence of these barriers was increased when it came to adherence to DOTS regimen.

*"Government has distributed TB medicines for free now [DOTS] but travel fare between my home and the health centre costs around Rs.500".*

*(FGD among seven community members, Nawalparasi)*

**Economic barrier.**  One of the prominent barriers that was often cross-cutting to all other barriers were the cost of attending health centre. Although, attending and accessing the health services were free of charge, the indirect cost associated with the travelling, staying in a hotel while waiting for the 3 daily sputum tests, food and accessories were prominent. These costs together with the opportunity costs related to losing the household chores, and agricultural works and if employed, the jobs were often pronounced. Such barriers were often appreciated not only by the patients but also by the health staff.

*"They have to stay in a hotel after coming here. They do not know that the sputum test including medication does not cost any money. However, they require money to travel from distant places and then they have to stay in a hotel. Sputum tests require a test for three consecutive days and thus diagnosis takes time and they have to bear the cost of waiting".*

*(IDI, Health staff, Kaski)*

Furthermore, the cost associated with visiting the health centre and with a companion from a long distance further impedes them from accessing the health centre and often patients were reported to visit when the disease was severe. Such a late visit also affected the early diagnosis and the chances of transmission to the family members and neighbours.

**Awareness about TB and traditional belief to access TB service.**  Although most of the respondents knew the name of the disease, in general, there was a lack of knowledge regarding the prominent symptoms of the disease such as low-grade fever, cough and sputum mixed with blood. Such a lack of knowledge further allowed them to seek alternative care rooted in the traditional belief. One of the first places where community members often visited as soon as they become ill were the traditional healers *[Dhami, Jhakri and Lama]*. Such a practice of visiting traditional healers are often rooted in the long-established belief that the illnesses are the results of deeds of past lives and karma. In response, these traditional healers are visited with the animals for sacrifice such as hen.

*"Villagers have a practice of taking the patient to a traditional healer, not to medical services in the beginning. People believe in these sorcerers when they are suffering from disease and readily sacrifice animals such as goat and hen. . .".*

*(FGD with eight TB suspected patients, a quote from a newly diagnosed TB patient, Kathmandu)*

Health staff were concerned that in general these patients only seek care at health centres after not being cured of medications they took from private pharmacies (who provide antibiotics without prescription) and traditional healers. Often, patients visited the health centre after failing to cure of multiple visits at various health places.

*"I had a small wound at my neck, I had pain since 1 month. I took medicine and somehow felt fine. Then I left taking medicine. And I again started to have pain after 2–4 days. . . I felt difficult to swallow then I came to a hospital. After taking antibiotics for 10 days, I went to another hospital. There also, they provided medicine for 10 days. Then I went to TB hospital in Dang [district] and I was diagnosed with TB. If my condition was diagnosed before then I could have taken medicine a month earlier".*

*(FGD with seven TB suspected patients, a quote from a patient who came for the treatment at Kathmandu)*

Patients also came up with the history of visiting multiple hospitals where they were treated as a patient of fever for a long time and this further delayed the diagnosis and referral.

Lack of knowledge about the TB in general further had grave consequences such as incomplete adherence to medication, specifically when they felt the relief of symptoms after taking medication for two months which may result in increased relapses and multidrug-resistant TB.

**Stigma and misconception.** Patients who are taking TB medications face a myriad of social challenges. They are often stigmatized based on the fear of transmission and misconception that the disease is due to karmic retributions. Concealing the status of diagnosis and treatment was often echoed by respondents who shared that (if the status is known to others), they could be asked to leave the rented room, asked to take a leave from the school or the job. Such a fear was rooted in past social exclusions and new TB patients thus seem to hide the disease.

*"People [TB patients] get scared. . .like people who stay in rent are here and if the house-owner comes to know that they have TB then the owner will ask them to leave his/her house. And if they will tell me to do so then where will I go?" This fear makes difficult for me to come to the health facility".*

*(FGD with six community members, Kaski)*

**Regimen complexity.** Visiting health centres for daily doses of TB was seen as a major impediment to treatment adherence. In addition to the stigma attached to the disease which made patients feel scared to visit health centres every day, the other factors that often barred them to visit the health centre for daily regimen were the distance, direct and indirect costs, and the patient's bodily conditions to travel. In addition, the long regimen for TB treatment and the apparent feeling of being cured after two months of treatment deterred patients from completing the whole regimen.

*"We have to come regularly to take TB medicines. This is difficult for children and elderly patients. How can we come every day to take medicine at the health centre? We have children at home, household works and for some people, they might be in daily wage. Some people may stop taking medicine in the middle".*

*(FGD with five Community members, Kathmandu)*

## Discussion

This study draws from a mixture of methods with diverse participants on the barriers to access, diagnosis and treatment completion for Tuberculosis from three geographic regions of Nepal. The barriers in this study are broadly from 1. Characteristics of the health system and 2. Care seeker's socio-demographic characteristics. Health system level barriers originated from poor functioning of peripheral health structure due to the irregular presence of (trained) human resources, scarcity of equipment (for diagnosis), and shortage of medication.

Care seekers' characteristics were prominent barriers due to the poor accessibility to the health centre (long distance, poor roads and trail conditions, lack of availability of public transport), economical constraints (direct and indirect costs associated with the travel). These barriers were further augmented by poor awareness regarding the TB, delayed treatment seeking, alternative visits for treatment seeking (for example traditional healers and on the counter medication), regimen complexity and stigma associated with the TB.

### Health system characteristics

Human resources are the critical elements for the functioning of peripheral health structures globally [27, 28] and more importantly in resource-constrained settings of Nepal [29–31]. As has been established previously, human resources act as the first interface between the patients and the peripheral health centres [27]. In Nepal, a serious scarcity of both trained and untrained human resources to ensure TB patients receive access, diagnosis and treatment is a major problem [20].

In the majority of instances, patients' visits at health centre are encountered by lack of staff and the lack of diagnostics. Sometimes, even with the presence of staff, poor communication to patients may lead to the perception of inadequate time with the health workers which can further lead to lack of comprehensibility (of the disease and the prescription), and thus may perpetuate the mistrust and poor adherence. Lack of adequate time and poor attitude by health service providers were major barriers in adherence to the treatment in Pakistan [32].

Globally, misdiagnosis and delayed diagnosis are affected by many other factors even within the settings of the Hospitals [33–35]. Delayed diagnosis can leave patients and their family members vulnerable to seek care from formal and informal health services and may take unnecessary and inappropriate doses of antimicrobials available over the counter in Nepal [36, 37]. This can further lead to additional delays, distortion of the clinical conditions and ultimately antimicrobial resistance [37]. Although few participants were explained and treated by auxiliary midwives, there were clear utterances that their dissatisfaction was mostly rooted in the misdiagnosis and its consequences. Such a practice can augment the continuation of infection at the family and village level [34]. Together with the incomplete follow-ups and adherence, patients may ultimately develop (multi-drug) resistant TB [38].

### Socio-demographic characteristics of patients

Socio-demographic characteristics of patients are often multi-burdened by their own limitations that include the accessibility to the health centres, economic constraints, patients' awareness and practice of visiting multiple health care providers. These factors are further complicated by the stigma attached to being diagnosed as TB patients, seeking treatment at the health centre including compliance for a long and complicated TB regimen [39].

Nepal's geographical landscape poses challenges in relation to access to the health centre and hospitals [40]. As has been established previously, longer distance and poor road (trail) conditions discourage patients to visit health centres. Patients face further hindrances due to lack of public transport and the cash money to afford such travel [41–43].

While these factors are already insurmountable in remote regions of Nepal, a further layer of barriers multiply to the poor attendance and health service utilization, that for example includes the stigma attached to being diagnosed as a TB patient [39]. Similar to disease conditions such as leprosy and HIV, diagnosis of TB and being seen as taking TB medications can lead to decrease social participation, perceived stigma and thus concealment of conditions among families, friends and the broader society [44–48]. A person may also suffer from both enacted and perceived form of courtesy stigma that may affect their family members negatively in marriage arrangements, occupation and social opportunities [49, 50].

Since the launch of DOTS (Directly Observed Treatment, short course) in 1995 by WHO, millions of patients worldwide have benefitted from it [51]. Nevertheless, DOTS strategy and its success are dependent upon the characteristics of the health system and the care seekers [52]. A long and complex regimen of TB treatment can discourage compliance, particularly when compounded by the patient's condition, accessibility, and economic constraints. A patient may forget, feel exhausted to take the medicine and more importantly may fall prey to the cycle of distance, money and time required for the current DOTS treatment that requires a daily visit to the health centre [52, 53].

Although there are only emerging evidence in recent years about how to compliment the DOTS strategy, a clear indication from this study is to review and strengthen the DOTS strategy through multiple measures. For instance, the daily DOTS regimen could be replaced by a monthly provision of medicine to be monitored by a family member or a locally available community member with a minimal counselling on how to take the medicine. Utilizing the elements of community engagement [54, 55], community members or family members can be asked to take a role in monitoring the daily intake of medication. For instance in HIV/AIDS, peer support (treatment buddies) for the compliance of medication has been found to be useful [56, 57]. Building on these, a wider level of engagement within the community to support the early treatment seeking and its benefits, and the compliance to the treatment regimen can become promising. Community engagement may further counteract the social stigma attached to TB, promote social participation and psychosocial well-being of the patients.

## Implications for health policy and planning to mitigate barriers to access, diagnosis and treatment completion

This study has highlighted barriers inherent in the current health system and broader social and cultural context of Nepal. Health system improvements for the functioning of peripheral health structures are critical. Although re-location of health structures and improvement in accessibility may cost a huge investment at least in the nearest future, a realistic approach would be to improve the functionality of existing peripheral health structures. The functionality can be promoted by 1. Adequately mobilizing the required human resources in the peripheral health structures and 2. Ensuring the provision of equipment for the diagnosis of TB and medications for TB treatment.

In regards to socio-demographic factors, an increase in awareness and health education to promote the recognition of TB, attendance at TB treatment centres are indispensable. This may further imply collaborating with both private sector health providers and traditional health providers [43]. With adequate collaboration with all kinds of health service providers, health service utilization may be enhanced. Nevertheless, this may not be enough to mitigate the barriers such as distance, poor road conditions, monetary constraints in attending the health centres. So, it may be pragmatic to identify community members as volunteers for the facilitation of DOTS supervision.

DOTS supervision can be further strengthened by training the family member/peer or neighbour who often accompany the TB patients. In case they do not accompany, patients could be asked to come along with such companions and can be easily trained to supervise the DOTS treatment at the village level.

Globally, community engagement has been increasingly promoted in the wider health programs such as malaria, leishmaniasis, river blindness and vitamin A supplementation [55]. Community volunteers can be trained and established as a representative [54] in remote regions of Nepal [58] where they can promote the early detection, diagnosis and treatment completion. Findings from this study can be tailored to mitigate the barriers such as difficulties of access, diagnosis and treatment completion.

## Conclusion

A range of barriers inherent in 1. Health services and 2. Socio-demographic characteristics of the care seekers can be dealt with by strengthening the peripheral health services. Specifically, health centres can be improved by ensuring the presence of (trained) human resources and necessary equipment for diagnosis. As well as increasing the awareness and collaborating with the traditional healers, health services utilization can be enhanced by compensating the direct and indirect costs associated with it including the modification in current DOTS strategy by providing medicine for longer term under the supervision of a family member or a (volunteer) community member utilizing the elements of community engagement.

## Acknowledgments

We are grateful to former directors of National Tuberculosis Centres: Dr. Kedar Narsing KC, Dr. Bikash Lamichhane and Dr. Rajendra Pant for the coordination of this study. We would like to thank all District (Public) Health Authorities for the facilitation of the study: Shambhu Prasad Gyawali (Mustang), Sagar Prasad Ghimire (Kaski), Durga Datta Chapagain (Tanahun), Shree Krishna Bhatta (Kathmandu), Jaya Bahadur Karki (Nawalparasi) and Raj Kishor Pandit (Parsa).

## Author Contributions

**Conceptualization:** Sujan Babu Marahatta, Ashish Shrestha, Pramod Raj Bhattrai, Bipin Adhikari.

**Data curation:** Sujan Babu Marahatta, Rajesh Kumar Yadav, Deena Giri, Sarina Lama, Komal Raj Rijal, Shiva Raj Mishra, Ashish Shrestha, Pramod Raj Bhattrai, Roshan Kumar Mahato, Bipin Adhikari.

**Formal analysis:** Sujan Babu Marahatta, Rajesh Kumar Yadav, Deena Giri, Sarina Lama, Komal Raj Rijal, Shiva Raj Mishra, Pramod Raj Bhattrai, Roshan Kumar Mahato, Bipin Adhikari.

**Funding acquisition:** Sujan Babu Marahatta, Komal Raj Rijal.

**Investigation:** Sujan Babu Marahatta, Rajesh Kumar Yadav, Deena Giri, Sarina Lama, Shiva Raj Mishra, Ashish Shrestha, Pramod Raj Bhattrai, Roshan Kumar Mahato, Bipin Adhikari.

**Methodology:** Sujan Babu Marahatta, Deena Giri, Sarina Lama, Komal Raj Rijal, Shiva Raj Mishra, Ashish Shrestha, Pramod Raj Bhattrai, Roshan Kumar Mahato, Bipin Adhikari.

**Project administration:** Sujan Babu Marahatta, Rajesh Kumar Yadav, Deena Giri, Sarina Lama, Shiva Raj Mishra, Ashish Shrestha, Pramod Raj Bhattrai, Roshan Kumar Mahato.

**Resources:** Sujan Babu Marahatta, Rajesh Kumar Yadav, Deena Giri, Sarina Lama, Komal Raj Rijal, Ashish Shrestha, Pramod Raj Bhattrai, Roshan Kumar Mahato, Bipin Adhikari.

**Software:** Sujan Babu Marahatta, Deena Giri, Sarina Lama, Shiva Raj Mishra, Ashish Shrestha, Roshan Kumar Mahato, Bipin Adhikari.

**Supervision:** Sujan Babu Marahatta, Rajesh Kumar Yadav, Deena Giri, Sarina Lama, Ashish Shrestha, Pramod Raj Bhattrai, Roshan Kumar Mahato, Bipin Adhikari.

**Validation:** Sujan Babu Marahatta, Rajesh Kumar Yadav, Deena Giri, Sarina Lama, Shiva Raj Mishra, Ashish Shrestha, Pramod Raj Bhattrai, Roshan Kumar Mahato, Bipin Adhikari.

**Visualization:** Sujan Babu Marahatta, Rajesh Kumar Yadav, Deena Giri, Sarina Lama, Ashish Shrestha, Pramod Raj Bhattrai, Roshan Kumar Mahato, Bipin Adhikari.

**Writing – original draft:** Sujan Babu Marahatta, Deena Giri, Sarina Lama, Komal Raj Rijal, Bipin Adhikari.

**Writing – review & editing:** Sujan Babu Marahatta, Shiva Raj Mishra, Ashish Shrestha, Bipin Adhikari.

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
