## [Decision Letter · Decision Letter 0]

10 Sep 2019

PONE-D-19-15886

Barriers in the access, diagnosis and treatment completion for Tuberculosis patients in central and western Nepal: a qualitative study among patients, community members and health care workers

PLOS ONE

Dear Dr. Adhikari,

Thank you for submitting your manuscript to PLOS ONE. After careful consideration, we feel that it has merit but does not fully meet PLOS ONE’s publication criteria as it currently stands. Therefore, we invite you to submit a revised version of the manuscript that addresses the points raised during the review process.

We would appreciate receiving your revised manuscript by Oct 18 2019 11:59PM. To enhance the reproducibility of your results, we recommend that if applicable you deposit your laboratory protocols in protocols.io, where a protocol can be assigned its own identifier (DOI) such that it can be cited independently in the future. For instructions see: http://journals.plos.org/plosone/s/submission-guidelines#loc-laboratory-protocols

We look forward to receiving your revised manuscript.

Kind regards,

Lars-Peter Kamolz, M.D., Ph.D., M.Sc.

Academic Editor

PLOS ONE

Journal Requirements:

1. We noticed you have some minor occurrence of overlapping text with the following previous publications, which needs to be addressed:

* Issues and threats of Tuberculosis in Nepal, by Amrit Banstola https://www.ghdonline.org/ic/discussion/issues-and-threats-of-tuberculosis-in-nepal/

* Waisbord, Silvio. "Behavioral barriers in tuberculosis control: A literature review." Washington, DC (2004).

* Naidoo, Kogieleum, et al. "Addressing challenges in scaling up TB and HIV treatment integration in rural primary healthcare clinics in South Africa (SUTHI): a cluster randomized controlled trial protocol." Implementation Science 12.1 (2017): 129.

In your revision ensure you cite all your sources (including your own works), and quote or rephrase any duplicated text outside the methods section. Further consideration is dependent on these concerns being addressed.

2. Please include a separate caption for each figure in your manuscript.

Reviewers' comments:

Reviewer's Responses to Questions

**Comments to the Author**

1. Is the manuscript technically sound, and do the data support the conclusions?

Reviewer #1: Yes

Reviewer #2: Yes

Reviewer #3: Yes

2. Has the statistical analysis been performed appropriately and rigorously? 

Reviewer #1: N/A

Reviewer #2: N/A

Reviewer #3: No

3. Have the authors made all data underlying the findings in their manuscript fully available?

Reviewer #1: Yes

Reviewer #2: Yes

Reviewer #3: Yes

4. Is the manuscript presented in an intelligible fashion and written in standard English?

Reviewer #1: Yes

Reviewer #2: Yes

Reviewer #3: Yes

5. Review Comments to the Author

Reviewer #1: interesting and well-written manuscript about a region with high burden of TB in the world and is involved in different problems and issues, a different experience versus developed countries

Reviewer #2: Tuberculosis is a common issue in developing countries. Studied shed light on operational problems are paramount to boost TB control.

Reviewer #3: The authors aim to explore the factors affecting the access to the health services, diagnosis and the treatment completion for TB patients using a qualitative method in central and western Nepal.

My suggestions for improvement are as follows:

GENERAL REPORTING

Please write the findings of your study in line with the reporting recommendations/checklist for qualitative studies; "Tong A SP, Craig J. Consolidated criteria for reporting qualitative research (COREQ): a 32-item checklist for interviews and focus groups. Int J Qual Health Care. 2007;19(6):349-57". Some of the recommended items have not been reported in this manuscript.

OBJECTIVE

-It would be good if the objective in the abstract matches that in the main text.

Abstract: to explore the factors affecting the access to the health services, diagnosis and the treatment completion for TB patients using a qualitative method in central and western Nepal.

Main text: to explore the barriers to access, diagnosis and treatment completion in all three ecological zones (various

geographic regions) within six districts of Nepal using a qualitative method.

-I don't think there is need to state 'using a qualitative method"in the objective. Stating that the aim was to explore implies a qualitative approach and the methods section clearly states that a qualitative approach was used.

ABSTRACT

-Results section: Please rephrase: "Early diagnosis of TB WERE hindered by lack of trained health personnel" to Early diagnosis of TB WAS hindered by lack of trained health personnel

-Conclusion: Please delete the numbers in the sentence: "Barriers embedded in 1. Health services and 2. Care seekers’ characteristics can be dealt by strengthening the peripheral health services"

METHODS

Study design and participants: "This was a qualitative cross-sectional study". Please delete the term cross sectional study. Cross-sectional study implies a quantitative approach was also used.

It is unclear what the authors mean by "the participants were considered adequate if reached theoretical saturation using standard qualitative methods". Please describe what is meant by "standard qualitative methods". The authors reference a paper on the grounded theory method. Grounded theory is one of the five approaches or methods used to guide the collection of qualitative data. Data saturation affects all types of approaches not only grounded theory.

Following above statement: It would be useful if readers state which qualitative approach guided the collection of data. Five methods have been proposed; Ethnography, Narrative, Phenomelogical, Grounded theory and Case study. Please visit this weblink for a summary of the differences between the approaches (https://measuringu.com/qual-methods/). From my understanding the phenomelogical approach applies to this study. Please clarify with appropriate references

ANALYSIS

What type of analysis was done? Thematic analysis/Content analysis/Framework analysis. Please clarify and write this section based on the recommended methods of qualitative analysis. Do reference as appropriate.

RESULTS

-The introductory paragraph on health system level barrier reads more like a discussion section than a results section. Please revise.

-The authors state that two main themes were explored; each with their various subthemes. It would be useful to present a summary table with themes and subthemes. The presentation of figure two does not adequately present the themes and subthemes discussed. From the figure it seems the main themes were: Health system characteristics; Characteristics of the regimen and Health seeker's characteristics.

6. PLOS authors have the option to publish the peer review history of their article (what does this mean?). If published, this will include your full peer review and any attached files.

Reviewer #1: Yes: Ilad Alavi Darazam

Reviewer #2: Yes: Layth Al-Salihi

Reviewer #3: Yes: Eleanor Ochodo

---

## [Author Response · Author response to Decision Letter 0]

16 Sep 2019

16th September, 2019

Dear editor and reviewers, 

We are very grateful to your suggestions, comments and relevant references. Your suggestions and comments were very helpful and we have revised the manuscript based on your suggestions. 

Below, we have added our responses with relevant corresponding changes in the manuscript. 

We look forward to your kind consideration. 

Sincerely yours, 

On behalf of co-authors, 

Bipin Adhikari

AUTHORS: Thank you for the suggestion. We have revised based on the formatting guidelines.

We noticed you have some minor occurrence of overlapping text with the following previous publications, which needs to be addressed:

* Issues and threats of Tuberculosis in Nepal, by Amrit Banstola https://www.ghdonline.org/ic/discussion/issues-and-threats-of-tuberculosis-in-nepal/

* Waisbord, Silvio. "Behavioral barriers in tuberculosis control: A literature review." Washington, DC (2004).

* Naidoo, Kogieleum, et al. "Addressing challenges in scaling up TB and HIV treatment integration in rural primary healthcare clinics in South Africa (SUTHI): a cluster randomized controlled trial protocol." Implementation Science 12.1 (2017): 129. 

In your revision ensure you cite all your sources (including your own works), and quote or rephrase any duplicated text outside the methods section. Further consideration is dependent on these concerns being addressed.

AUTHORS: Thank you for these suggestions. We have revised the paragraphs and added the relevant references. 

2. Please include a separate caption for each figure in your manuscript.

Reviewers' comments:

Reviewer's Responses to Questions

Comments to the Author

1. Is the manuscript technically sound, and do the data support the conclusions?

Reviewer #1: Yes

Reviewer #2: Yes

Reviewer #3: Yes 

2. Has the statistical analysis been performed appropriately and rigorously?

Reviewer #1: N/A

Reviewer #2: N/A

Reviewer #3: No 

3. Have the authors made all data underlying the findings in their manuscript fully available?

Reviewer #1: Yes

Reviewer #2: Yes

Reviewer #3: Yes

4. Is the manuscript presented in an intelligible fashion and written in standard English?

Reviewer #1: Yes

Reviewer #2: Yes

Reviewer #3: Yes

5. Review Comments to the Author

Reviewer #1: interesting and well-written manuscript about a region with high burden of TB in the world and is involved in different problems and issues, a different experience versus developed countries

Reviewer #2: Tuberculosis is a common issue in developing countries. Studied shed light on operational problems are paramount to boost TB control.

Reviewer #3: The authors aim to explore the factors affecting the access to the health services, diagnosis and the treatment completion for TB patients using a qualitative method in central and western Nepal.

My suggestions for improvement are as follows:

GENERAL REPORTING

Please write the findings of your study in line with the reporting recommendations/checklist for qualitative studies; "Tong A SP, Craig J. Consolidated criteria for reporting qualitative research (COREQ): a 32-item checklist for interviews and focus groups. Int J Qual Health Care. 2007;19(6):349-57". Some of the recommended items have not been reported in this manuscript.

AUTHORS: Thank you very much for the useful recommendation. We have revised the methods section in relevant areas to include the items included in the COREQ. 

OBJECTIVE

-It would be good if the objective in the abstract matches that in the main text.

Abstract: to explore the factors affecting the access to the health services, diagnosis and the treatment completion for TB patients using a qualitative method in central and western Nepal.

Main text: to explore the barriers to access, diagnosis and treatment completion in all three ecological zones (various

geographic regions) within six districts of Nepal using a qualitative method.

-I don't think there is need to state 'using a qualitative method"in the objective. Stating that the aim was to explore implies a qualitative approach and the methods section clearly states that a qualitative approach was used.

AUTHORS: Thank you for the suggestion. We have revised and retained the concise objective in both abstract and main text as suggested. 

ABSTRACT

-Results section: Please rephrase: "Early diagnosis of TB WERE hindered by lack of trained health personnel" to Early diagnosis of TB WAS hindered by lack of trained health personnel

AUTHORS: Thank you for the suggestion. Revised as suggested. 

-Conclusion: Please delete the numbers in the sentence: "Barriers embedded in 1. Health services and 2. Care seekers’ characteristics can be dealt by strengthening the peripheral health services"

AUTHORS: Thank you for the suggestion. Revised as suggested. 

METHODS

Study design and participants: "This was a qualitative cross-sectional study". Please delete the term cross sectional study. Cross-sectional study implies a quantitative approach was also used.

AUTHORS: Thank you for the suggestion. Revised as suggested.

It is unclear what the authors mean by "the participants were considered adequate if reached theoretical saturation using standard qualitative methods". Please describe what is meant by "standard qualitative methods". The authors reference a paper on the grounded theory method. Grounded theory is one of the five approaches or methods used to guide the collection of qualitative data. Data saturation affects all types of approaches not only grounded theory.

AUTHORS: Thank you for the suggestion. We have revised the sentence to ensure it is understood. We also agree that the reference we have included expounds grounded theory method and thus may misguide readers that the concept of theoretical saturation only applies to such method (in contrast to all qualitative methods). Nevertheless, (following a tradition in reporting) we chose this reference because Glaser and Strauss were the first to discuss the concept of theoretical saturation in 1967 in the very book and the concept was subsequently taken up in other qualitative methods. In light of this and to further clarify, we have added a reference: a recent discourse (Saunders 2018: Saturation in qualitative research: exploring its conceptualization and operationalization) on the concept of theoretical saturation by Benjamin Saunders and colleagues which further substantiates the concept and its origin. 

Following above statement: It would be useful if readers state which qualitative approach guided the collection of data. Five methods have been proposed; Ethnography, Narrative, Phenomelogical, Grounded theory and Case study. Please visit this weblink for a summary of the differences between the approaches (https://measuringu.com/qual-methods/). From my understanding the phenomelogical approach applies to this study. Please clarify with appropriate references

AUTHORS: Thank you very much for the suggestion and the link for the qualitative research methods. We agree with you that this study utilized phenomenological approach in qualitative research. We have revised with an appropriate reference under the section ‘Study design and participants’.

ANALYSIS

What type of analysis was done? Thematic analysis/Content analysis/Framework analysis. Please clarify and write this section based on the recommended methods of qualitative analysis. Do reference as appropriate.

AUTHORS: Thank you very much for the suggestion. The study utilized a thematic analysis and broadly, themes were categorized into health system level barriers and care seekers’ socio-demographic barriers. We have revised the methods section to explicitly mention the thematic analysis conducted in this study. 

RESULTS

-The introductory paragraph on health system level barrier reads more like a discussion section than a results section. Please revise.

AUTHORS: Thank you for the suggestion. We have revised the paragraph and reads as follows: 

A: Health System Level Barrier 

Nepal has a unique landscape with regions divided between the northern Himalayas bordering China, mid mountains and the southern plains bordering India (Figure 1). Nepal has established a good network of peripheral health structures over the decades which serves the rural population. However, this study identified myriad of factors that affected the health service utilization leading to a neglect of one of the killer disease: Tuberculosis (Figure 2). Below, we outline each of these factors affecting health service utilization.

-The authors state that two main themes were explored; each with their various subthemes. It would be useful to present a summary table with themes and subthemes. The presentation of figure two does not adequately present the themes and subthemes discussed. From the figure it seems the main themes were: Health system characteristics; Characteristics of the regimen and Health seeker's characteristics.

AUHTORS: Thank you for this useful suggestion. Currently, in the manuscript, almost like a summary, we have presented each of the main two themes and the sub-themes in headings. One of the main issue with presenting them in table is: either the list of themes and sub-themes will just serve as a list of words/texts or if we opt to explain in tables, we will repeat the whole result section. Thus, following the conventional methods of presenting qualitative data with themes, sub-themes and the quotations, we believe, this will enhance readability. 

Thank you for your suggestion. Concerning figure 2, we agree with you that while the study classifies themes under ‘health system’ and health seekers’ characteristics as the main outcome-themes, ‘regimen complexity’ apparently reads as if like an independent theme. We had this discussion amongst authors as well and we decided that this will fall under health seekers’ characteristics (as presented in the result section of the manuscript). Nevertheless, regimen complexity permeates through both of the main themes: health system and care seekers’ characteristics as an interaction and interface between health system and the care seekers. Thus, we have revised figure 2 to show an inter-link between health system and the care seekers’ characteristics. Also, one of our main goal is to ensure that the figure adequately summarizes the results without being too complex, thus making it simple and readable for policy makers and international readers. We believe that the revised figure is as simple as we could produce.

---

## [Decision Letter · Decision Letter 1]

4 Dec 2019

PONE-D-19-15886R1

Barriers in the access, diagnosis and treatment completion for Tuberculosis patients in central and western Nepal: a qualitative study among patients, community members and health care workers

PLOS ONE

Dear Dr. Adhikari,

Thank you for submitting your manuscript to PLOS ONE. After careful consideration, we feel that it has merit but does not fully meet PLOS ONE’s publication criteria as it currently stands. Therefore, we invite you to submit a revised version of the manuscript that addresses the points raised during the review process.

We would appreciate receiving your revised manuscript by Jan 18 2020 11:59PM. To enhance the reproducibility of your results, we recommend that if applicable you deposit your laboratory protocols in protocols.io, where a protocol can be assigned its own identifier (DOI) such that it can be cited independently in the future. For instructions see: http://journals.plos.org/plosone/s/submission-guidelines#loc-laboratory-protocols

We look forward to receiving your revised manuscript.

Kind regards,

Lars-Peter Kamolz, M.D., Ph.D., M.Sc.

Academic Editor

PLOS ONE

Reviewers' comments:

Reviewer's Responses to Questions

**Comments to the Author**

1. If the authors have adequately addressed your comments raised in a previous round of review and you feel that this manuscript is now acceptable for publication, you may indicate that here to bypass the “Comments to the Author” section, enter your conflict of interest statement in the “Confidential to Editor” section, and submit your "Accept" recommendation.

Reviewer #2: All comments have been addressed

Reviewer #4: (No Response)

2. Is the manuscript technically sound, and do the data support the conclusions?

Reviewer #2: Yes

Reviewer #4: Yes

3. Has the statistical analysis been performed appropriately and rigorously? 

Reviewer #2: Yes

Reviewer #4: Yes

4. Have the authors made all data underlying the findings in their manuscript fully available?

Reviewer #2: Yes

Reviewer #4: Yes

5. Is the manuscript presented in an intelligible fashion and written in standard English?

Reviewer #2: Yes

Reviewer #4: Yes

6. Review Comments to the Author

Reviewer #2: (No Response)

Reviewer #4: Dear authors,

thank you for the opportunity to re-review the mansucript „Barriers in the access, diagnosis and treatment completion for Tuberculosis patients in central and western Nepal: a qualitative study among patients, community members and health care workers”.

I have only a small note:

Please delete the information on gender and age in the in-vivo codes.

7. PLOS authors have the option to publish the peer review history of their article (what does this mean?). If published, this will include your full peer review and any attached files.

Reviewer #2: Yes: Layth Al-Salihi

Reviewer #4: No

---

## [Author Response · Author response to Decision Letter 1]

4 Dec 2019

5th December 2019

Dear editor and reviewers, 

We are very grateful for your evaluation of the manuscript. We noticed that you had a small note/recommendation for us. 

Below, we have added our response with relevant corresponding changes in the manuscript. 

We look forward to your kind consideration. 

Sincerely yours, 

On behalf of co-authors, 

Bipin Adhikari

Comments to the Author

Reviewer #4: Dear authors,

thank you for the opportunity to re-review the mansucript „Barriers in the access, diagnosis and treatment completion for Tuberculosis patients in central and western Nepal: a qualitative study among patients, community members and health care workers”.

I have only a small note:

Please delete the information on gender and age in the in-vivo codes.

AUTHORS: Thank you very much for this suggestion. We have removed the information on age and gender below each quote within the revised manuscript.

---

## [Decision Letter · Decision Letter 2]

17 Dec 2019

Barriers in the access, diagnosis and treatment completion for Tuberculosis patients in central and western Nepal: a qualitative study among patients, community members and health care workers

PONE-D-19-15886R2

Dear Dr. Adhikari,

We are pleased to inform you that your manuscript has been judged scientifically suitable for publication and will be formally accepted for publication once it complies with all outstanding technical requirements.

With kind regards,

Lars-Peter Kamolz, M.D., Ph.D., M.Sc.

Academic Editor

PLOS ONE

Additional Editor Comments (optional):

Reviewers' comments:

Reviewer's Responses to Questions

**Comments to the Author**

1. If the authors have adequately addressed your comments raised in a previous round of review and you feel that this manuscript is now acceptable for publication, you may indicate that here to bypass the “Comments to the Author” section, enter your conflict of interest statement in the “Confidential to Editor” section, and submit your "Accept" recommendation.

Reviewer #2: All comments have been addressed

Reviewer #3: All comments have been addressed

2. Is the manuscript technically sound, and do the data support the conclusions?

Reviewer #2: Yes

Reviewer #3: Yes

3. Has the statistical analysis been performed appropriately and rigorously? 

Reviewer #2: Yes

Reviewer #3: Yes

4. Have the authors made all data underlying the findings in their manuscript fully available?

Reviewer #2: Yes

Reviewer #3: Yes

5. Is the manuscript presented in an intelligible fashion and written in standard English?

Reviewer #2: Yes

Reviewer #3: Yes

6. Review Comments to the Author

Reviewer #2: (No Response)

Reviewer #3: I have no further comments on this manuscript that addresses "Barriers in the access, diagnosis and treatment completion for Tuberculosis patients in central and western Nepal: a qualitative study among patients, community members and health care workers "

7. PLOS authors have the option to publish the peer review history of their article (what does this mean?). If published, this will include your full peer review and any attached files.

Reviewer #2: Yes: Layth Al-Salihi

Reviewer #3: No

---

## [Editor Report · Acceptance letter]

19 Dec 2019

PONE-D-19-15886R2 

Barriers in the access, diagnosis and treatment completion for Tuberculosis patients in central and western Nepal: a qualitative study among patients, community members and health care workers 

Dear Dr. Adhikari:

I am pleased to inform you that your manuscript has been deemed suitable for publication in PLOS ONE. Congratulations! Your manuscript is now with our production department. 

With kind regards,

on behalf of

Dr. Lars-Peter Kamolz 

Academic Editor

PLOS ONE